# Pressure-Induced Variation of the Crystal Stacking Order in the Hydrogen-Bonded Quasi-Two-Dimensional Layered Material Cu(OH)Cl

**DOI:** 10.3390/ma14175019

**Published:** 2021-09-02

**Authors:** Hui Tian, Meiling Wang, Jian Zhang, Yanmei Ma, Hang Cui, Jiaxin Zhao, Qing Dong, Qiliang Cui, Bingbing Liu

**Affiliations:** 1State Key Laboratory of Superhard Materials, College of Physics, Jilin University, Changchun 130012, China; tianhui18@mails.jlu.edu.cn (H.T.); mlwang20@mails.jlu.edu.cn (M.W.); ymma@jlu.edu.cn (Y.M.); cuihang@jlu.edu.cn (H.C.); cql@jlu.edu.cn (Q.C.); liubb@jlu.edu.cn (B.L.); 2Alan G. MacDiarmid Institute, College of Chemistry, Jilin University, Changchun 130012, China; zhaojx19@mails.jlu.edu.cn

**Keywords:** high pressure, Cu(OH)Cl, 2D layered materials, crystal stacking order

## Abstract

The crystal stacking order plays a crucial role in determining the structure and physical properties of 2D layered materials. A variation in the stacking sequence of adjacent 2D building blocks causes drastic changes in their functionalities. In this work, the structural variation of belloite (Cu(OH)Cl), as a function of pressure, is presented. Through in situ synchrotron X-ray diffraction and Raman scattering studies, in combination with first-principles theoretical simulations, a structural transformation from the initial monoclinic phase into an orthorhombic one has been established at 18.7 GPa, featuring variations in the stacking sequence of the tectonic monolayers. In the monoclinic phase, they are arranged in an AAAA sequence. While in the orthorhombic phase, the monolayers are stacked in an ABAB sequence. Such phenomena are similar to those observed in van der Waals 2D materials, with pressure-induced changes in the stacking order between layers. In addition, an isostructural phase transition within the initial monoclinic phase is also observed to occur at 12.9–16 GPa, which is associated with layer-sliding and a change in hydrogen bond configuration. These results show that Cu(OH)Cl, as well as other hydrogen-bonded 2D layered materials, can provide a convenient platform for studying the effects of the crystal stacking order.

## 1. Introduction

The effects of crystal stacking order on the physics of low-dimensional van der Waals (vdW) materials have attracted considerable current research enthusiasm [1,2]. Drastic changes in certain physical properties may be induced by even a slight variation in the relative position between adjacent one-dimensional or two-dimensional building blocks. For example, in two-dimensional van der Waals magnets of atomically thin CrI_3_ sheets, experiments and theoretical calculations show that interlayer exchange coupling is strongly dependent on layer separation, while the stacking arrangement can even change the sign of the interlayer magnetic exchange, thus drastically modifying the ground state [3,4,5,6,7,8,9]. Rhombohedral graphite (RG), a metastable polymorph of carbon, possesses a peculiar ABCABC stacking sequence of the graphene layers, in contrast to the ABAB stacking sequence in ordinary hexagonal graphite. Such a variation in the stacking sequence endows multi-layered rhombohedral graphite with gapped bulk electron states and electron transport properties that are dominated by topological surface states, which are distinct from those of its hexagonal counterpart [10]. In addition, in the quasi-one-dimensional van der Waals material of Bi_4_Br_4_, a slight modification of the stacking structure of the Bi_4_Br_4_ chains induces a transition from a trivial insulator to a higher-order topological insulator [11]. Another current focus of intense research activity concerns stacked two-dimensional crystals, correlated with symmetry [12]. The engineering of symmetry breaking can be effectively achieved by controlling the stacking order as well as the crystal phase. For example, for bilayer transition metal dichalcogenides, 3R stacking lacks inversion symmetry, while 2H stacking does have inversion symmetry. Because of the broken inversion symmetry, bilayer transition metal dichalcogenides with 3R stacking can show exotic valley physics and nonlinear optical responses [13,14,15,16,17]. These scientific findings suggest that the crystal stacking order has profound effects on the intrinsic nature of low-dimensional van der Waals materials. Control and regulation of the crystal stacking structures offer the opportunity to pave the path for progress in the science and technology of low-dimensional materials.

The modern high-pressure technique has been proven to be a clean, reversible, and versatile tool for the continuous control of interlayer coupling via interlayer spacing in vdW low-dimensional materials and bulk crystals. Hydrostatic pressure has recently been employed to modify the bands in graphene/hexagonal boron nitride (hBN) moiré superlattices [18] and transition metal dichalcogenides [19,20], as well as the correlated electronic phases in twisted bilayer graphene [21]. In particular, it was reported in a previous study that the irreversible antiferromagnetic-ferromagnetic transition between layers in atomically thin CrI_3_ is observed when hydrostatic pressure is applied to change the stacking order in the van der Waals magnetic insulator CrI_3_ [22,23].

As a necessary complement to vdW low-dimensional materials and bulk crystals when approaching the physical nature of the effects of crystal stacking structures, quasi-two-dimensional layered materials with interlayer bonding other than van der Waals interactions are a matter of interest. Layered metal hydroxyhalide/oxide, such as Cu(OH)F and γ-AlOOH [24,25], features a quasi-two-dimensional structure consisting of thin atomic sheets held together with weak-interlayer hydrogen bonding. The complicated compression behaviors of these MOHX (M is a metal, such as Cu, Zn, and Al, and X is for halogen or oxygen) materials, endowed by the collaboration and competition between the strong intralayer chemical bonds and weak interlayer hydrogen bond, have been revealed recently. MOHX materials comprise a large family of layered metal hydroxyhalide/oxide with diverse intralayer elemental compositions and interlayer hydrogen-bonding geometry. From this viewpoint, they provide a convenient platform for studying the effects of crystal stacking structures, as well as the coupling between interlayer hydrogen bonds and intralayer strong chemical bonds.

In this study, the pressure-induced variation of the crystal stacking order in the hydrogen-bonded quasi-two-dimensional layered metal hydroxychloride, Cu(OH)Cl, has been investigated systematically through experimental observations and theoretical simulations in parallel. The results herein are expected to contribute to a deeper insight into the physics related to the crystal stacking structure in quasi-two-dimensional materials in general.

## 2. Materials and Methods

### 2.1. Sample Preparation

All the chemicals involved are reagents of analytical grade and are used without further purification. In a typical synthetic process, powders of 5.68 g copper chloride dihydrate (CuCl_2_·2H_2_O) (Alfa Aesar, Shanghai, China) and 4.26 g copper oxide (CuO) (Alfa Aesar, Shanghai, China) were weighed proportionally. The raw materials were mixed and ground in an agate mortar for 2 h until a homogeneous mixture was obtained. The mixture was transferred into a 30 mL Teflon-lined autoclave. After being sealed and heated at 180 °C for 12 h, the autoclave was gradually cooled to room temperature. The mixtures in the autoclave were collected into an agate mortar and ground for an additional 30 min to make them homogeneous. They were then heat-treated again in the autoclave at the same temperature for the same time period. These processes were repeated several times, with a total grinding time of 5 h and a total reaction time of 60 h. Subsequently, the dark green powders of copper hydroxychloride, Cu(OH)Cl, were obtained.

### 2.2. In Situ High-Pressure XRD, Absorption and Raman Measurements

High-pressure experiments were carried out with a symmetric diamond anvil cell (DAC). Diamond anvils with culets of 300 μm diameter were used for this study. The sample and a small ruby ball were loaded into the 80 μm diameter chamber of a DAC, constructed from a T301 steel gasket pre-indented to a thickness of 45 μm. The pressure calibration was determined by utilizing the standard ruby fluorescent technique [26]. We performed in situ high-pressure angle-dispersive X-ray diffraction (ADXRD) measurements, with wavelengths of 0.6199 Å at 4W2 beamline, at the High-Pressure Station of the Beijing Synchrotron Radiation Facility (Beijing, China). The two-dimensional Debye–Scherrer diffraction rings were recorded using an imaging plate detector and were integrated into the one-dimensional profile using the Fit2D program [27]. Absorption spectra were measured by using a deuterium-halogen light source, and the measurements were between 250 nm and 1000 nm. The intercept of the absorption edge onto the energy axis from a plot of (αhν)^1/2^ versus photon energy (hν) gives a good approximation of the Cu(OH)Cl bandgap energy. High-pressure Raman measurements were performed using a LabRam HR Evolution spectrometer (Horiba, France) equipped with a 473 nm laser. In order to avoid the probable damage from heating or oxidizing under laser exposure, the laser power at the surface of the sample was attenuated to be less than 0.5 mW. The signals were dispersed by an 1800 g mm^−1^ (Raman) grating and were collected via a 20× objective lens under atmospheric conditions and high pressures. In the high-pressure experiments, silicon oil was utilized as the pressure-transmitting medium (PTM) for optical absorption and ADXRD experiments. For the Raman experiments, nitrogen served as the pressure transmitting medium.

### 2.3. Theoretical Simulations

All calculations were performed by using the first-principles method within the density functional theory (DFT) framework, as implemented in the Vienna ab initio simulation package (VASP) [28]. The projector augmented-wave (PAW) method employed with Perdew–Burke–Ernzerhof (PBE) [29] exchange-correlation functions are used to describe the interaction between valence electrons and ions. The plane wave energy cutoff and k-point sampling are set to be 550 eV and a 5 × 4 × 4 grid within the Monkhorst–Pack scheme [30], respectively, and the convergence threshold is set to be 1 × 10^−6^ eV/atom in energy and 1 × 10^−3^ eV Å^−1^ in force. The generalized gradient approximation (GGA) and the Hubbard correction U = 8 eV within the GGA+U scheme [31] was used to correct the electron delocalization that occurs in strongly correlated systems. The optimized lattice parameters for Cu(OH)Cl within GGA + U are a = 5.62 Å; b = 6.67 Å; c = 6.13 Å; β = 113.53°, which also matches well with reported experimental data [32]. The phonon spectrums are calculated via a combination of VASP (Version 5.4.4., Wien, Austria) and Phonopy (2009, Atsushi Togo) software [33].

## 3. Results

The XRD analysis was performed using an X-ray source of Cu Kα radiation (λ = 1.5406 Å). The typical XRD patterns of the prepared samples are shown in Figure 1a. All the detected peaks can be indexed to the monoclinic crystal structure (PDF no. 77-0324 P2_1_/a). No additional crystalline impurity phases can be observed, indicating the high purity of the obtained samples. The sharp peaks also indicate that the prepared samples have perfect crystallinity. In addition to XRD results, the chemical composition of the prepared samples was further checked by energy dispersive X-ray analysis (EDS). A typical EDS spectrum is shown in Figure 1b. It can be seen that the prepared sample contained mainly Cu, O and Cl elements. The Si element coming from the supporting substrate was also detected. According to the quantitative analysis, based on EDS, the Cu:O:Cl ratio was determined to be 0.95:0.98:1, indicating the nearly perfect stoichiometry of Cu(OH)Cl. Therefore, both XRD and EDS analyses show that pure monoclinic phase Cu(OH)Cl with high crystallinity was successfully synthesized by the present synthetic strategy. The morphological features of the prepared Cu(OH)Cl samples were observed via SEM (FEI Magellan 400L XHR, Tokyo, Japan) and TEM (JEOL, JEM-2200FS, Tokyo, Japan) techniques. The scanning electron microscope (SEM) image of the prepared sample, as shown in Figure 1c, reveals that the product is composed mostly of particles with diameters of 1–2 μm. Figure 1d shows a typical TEM micrograph of the prepared Cu(OH)Cl nanoparticles. It indicates that the particles have a thin lamella morphology, as observed at the overlapping region of two particles.

High-pressure powder X-ray diffraction (XRD) measurements of Cu(OH)Cl were performed under pressures of up to 28.4 GPa. As shown in Figure 2a, the XRD patterns at low pressures (e.g., 1.5 GPa) can be closely indexed to the monoclinic phase in space group P2_1_/a. All the diffraction peaks shift slightly to higher angles with increasing pressure, due to the shrinkage of the lattice. With the increase in pressure, the intensities of the diffraction peaks gradually weaken, and some weak peaks eventually disappear. As the pressure increased from 14.2 to 17 GPa, the Bragg peaks of (1 1 2) (see Figure 2b) and (1 2 −2) shifted notably to lower angles, implying that the d-spacings of (1 1 2) and (1 2 −2) abnormally increased upon compression. When the pressure reaches about 18.7 GPa, the characteristics of the diffraction pattern change significantly.

Two new diffraction peaks appeared at 7.1° and 12.7° (marked with arrows in Figure 2a), respectively. Figure 2c shows the two-dimensional XRD image of Cu(OH)Cl at 18.7 GPa. Two new diffraction rings can be observed, which further confirm that the appearance of the new peaks is indeed caused by structural changes. As the pressure increased, the intensities of the new peaks gradually increased, and they were maintained up to the highest pressure achieved in this study.

The pressure dependencies of the interplanar d-spacings and the lattice parameters of Cu(OH)Cl are demonstrated in Figure 3. It can be seen from Figure 3a that several d-spacings exhibit noteworthy anomalies in the variations of pressure at the critical points. The d-spacing of (1 1 −2) decreases monotonously below 14.2 GPa but remains nearly pressure-independent in the range of 14.2–17.5 GPa. The d-spacings of (1 1 2) and (1 2 −2) exhibited more exotic behavior. They decreased at low pressures; however, a peculiar increase occurred in their variation with increasing pressure. Considering the above results in combination, it can be determined that the structure of the initial phase (Phase-I) starts to transform above 14.2 GPa, and a new phase (Phase-II) is established at 18.7 GPa. After the phase transformation occurred, it was also observed that the (0 0 1), (1 1 −2) and (2 0 0) diffraction peaks, which are characteristic of phase-I, retain up to the highest pressure achieved in this study, with a continuous decrease in the values of their d-spacings with increased pressure. This means that the initial phase (Phase-I) may be preserved at high pressures, though it loses stability from a thermodynamic viewpoint. The pressure-dependency of the lattice parameters of Phase-I was ascertained from the XRD patterns and is shown in Figure 3b,f. The lattice parameter b decreased monotonously with increasing pressure. However, inflection points may be observed at 14.2 GPa in the plots of the lattice parameters a and c with pressure. The variation of the beta angle with pressure showed a discontinuity near 12.9 GPa. In order to show explicitly the subtle changes in compression behavior, the variations in the axial ratios of b/a and c/a with pressure were carefully checked and are plotted in Figure 3d,e, respectively. The ratio of b/a remained nearly constant at the first few gigapascals, and then increased rapidly to a maximum of 1.202 at 12.9 GPa, after which it decreased with the further increase in pressure. The ratio of c/a increased rapidly to a maximum at 14.2 GPa, and then remained almost constant with the further increase in pressure. These subtle changes in the lattice parameters were related to the changes in the hydrogen bonding configuration, as discussed below. The above results indicate that the initial structure of Cu(OH)Cl undergoes isostructural changes in the pressure range of 12.9–14.2 GPa.

In addition, the pressure dependency of the cell volume of Phase-I was fitted to the third-order Birch−Murnaghan (BM) equation of state (EOS). The fitting yielded the bulk modulus B_0_ = 49.2 GPa and its derivative B’_0_ = 10.2, as shown in Figure 3c. The relatively large value of B’ suggests a strong change of volume compressibility under pressure. The relatively lower bulk modulus of Cu(OH)Cl compared to Cu(OH)F (B_0_ = 61.1 GPa) suggests that Cu(OH)Cl is relatively easier to compress [24]. The reason lies in the fact that the interlayer hydrogen bonds in Cu(OH)Cl are weaker than those in Cu(OH)F.

In order to determine the structure of the new high-pressure phase (Phase-II), the XRD pattern of Cu(OH)Cl at 20 GPa was theoretically simulated (Figure 4a). An orthorhombic lattice structure with the space group Pbn2_1_ was considered as a proper candidate for Phase-II. It has been postulated previously by Voronova et al. [34] that such an orthorhombic (Pbn2_1_) structure of Cu(OH)Cl may exist, at least metastably. The two simulated patterns, according to the space groups P2_1_/a and Pbn2_1_, respectively, resembled each other to a large extent. They reflected similar configurations of atoms in the two phases under consideration. Notably, a plain superposition of the simulated XRD patterns of the two phases showed that the main features are in good agreement with the experimental XRD patterns at pressures above 18.7 GPa. To check the dynamical stability of the orthorhombic phase at high pressure, we calculated the phonon spectrum of Pbn2_1_ at 20 GPa. There were no imaginary frequencies appearing in the phonon dispersion curves (Figure 4b), confirming that the orthorhombic phase is dynamically stable at high pressures. To further test the validity of the postulated orthorhombic phase and to determine the transition pressure at T = 0 K, the enthalpies of the two phases were considered theoretically. For a given pressure, a stable structure is one with a lower value of enthalpy. Figure 4c shows the plots of our calculated enthalpies as a function of pressure for both monoclinic (P2_1_/a) and orthorhombic (Pbn2_1_) structures. It indicates that the energies of the two structures are very close to each other. The coexistence of the two phases may also be implied by their similarity in energy. The calculated relative enthalpy *ΔH* indicates that Cu(OH)Cl transforms into the orthorhombic structure at around 18 GPa (Figure 4d). These results further confirm that Cu(OH)Cl undergoes a phase transition from the monoclinic to the orthorhombic structure at 18.7 GPa. As pressure is further increased, the monoclinic and the orthorhombic phases coexist over a wide pressure range from 18.7 to at least 28.4 GPa. When the pressure is fully released, an obvious shoulder peak at about 6.6° is observed, accompanying the (0 0 1) diffraction peak of the initial Phase-I at 6.4°, which can be indexed to the (0 0 2) diffraction of phase-II (see Figure 2d). The diffraction peaks of phase-I show significant broadening and partial disappearance. Therefore, it is indicated that the novel high-pressure phase (Phase-II) may be recovered on returning to the ambient conditions.

In addition to the in situ high-pressure XRD studies, systematic in situ high-pressure Raman scattering measurements were conducted. Cu(OH)Cl samples were pressurized to a maximum of 34.1 GPa and the spectra were recorded upon both compression and decompression (Figure 5). According to symmetry analysis, the initial Cu(OH)Cl crystal belonged to the space group P2_1_/a (No. 14) and point group C^5^_2h_. The unit cell contains 16 atoms, hence there are 48 normal phonon modes at the Γ point:Γ = 12A_g_ (R) + 11A_u_ (IR) + 12B_g_ (R) + 10B_u_ (IR) + A_u_ + 2B_u_(1)
where A_g_ and B_g_ modes are Raman-active, and Au and Bu modes are infrared-active (IR). Therefore, there are 24 Raman-active (R) modes (Γ_Raman_ = 12A_g_ + 12B_g_), 21 infrared-active (IR) modes (Γ_IR_ = 11A_u_ + 10B_u_), and 3 acoustic modes (Γ_Acoustic_ = A_u_ + 2B_u_).

The Raman spectrum that was observed at ambient pressure is depicted in Figure 6. A good agreement between the theoretical and the experimental results was found, and each of the Raman peaks could be assigned successfully to the underlying vibrational modes. Table 1 summarizes the frequencies of the experimental and theoretical vibrational modes at Γ in Cu(OH)Cl at room pressure. The 24 Raman frequencies were obtained from the calculated phonon dispersion at the Γ point. Seventeen Raman modes were observed in the Raman scattering spectra at ambient pressure. A very good agreement was reached between the experimental and the calculated results, providing a direct assignment of the experimentally observed Raman shifts, based on their frequencies. All of them could be assigned to the monoclinic structure according to group analysis. The low-pressure frequencies are denoted as follows: 1A_g_ (72 cm^−1^), 2B_g_ (91 cm^−1^), 3A_g_ (109 cm^−1^), 4B_g_ (127 cm^−1^), 5A_g_ (137 cm^−1^), 6A_g_ (158 cm^−1^), 7B_g_ + 8B_g_ (171 cm^−1^), 9A_g_ (186 cm^−1^), 12B_g_ (247 cm^−1^), 13A_g_ (324 cm^−1^), 16B_g_ (391 cm^−1^), 17A_g_ + 18B_g_ (481 cm^−1^), 19A_g_ (807 cm^−1^), 20B_g_ (856 cm^−1^), 21A_g_ (870 cm^−1^), 22B_g_ (897 cm^−1^) and 23B_g_ + 24A_g_ (3388 cm^−1^).

The atomic displacements associated with the Raman-active modes in the primitive unit cells can be found in previous work [24]. The eighteen bands below 500 cm^−1^ are assigned to the lattice modes involving the deformation or relative displacement of the Cu(OH)_3_Cl_3_ octahedra. The four bands observed at 800–900 cm^−1^ correspond to the O–H bending modes. In the high-frequency region, two bands were observed at 3458 and 3460 cm^−1^, which could be assigned to the O–H stretching modes. At 1.9 GPa, a new Raman mode at 290 cm^−1^ (M_1_) appeared and increased in intensity with increasing pressure, up to about 27.6 GPa. The frequency of mode M_1_ is equal to one-half of the sum of the frequencies of 11A_g_ and 13A_g_ modes. It is indicated that the new Raman mode M1 may likely be due to the plasmon-phonon coupling band (L^+^ or L^−^) of the 11A_g_ and 13A_g_ modes. Under pressures below about 18.0 GPa, the O–H stretching mode went through a redshift with the increase in pressure. When the pressure was elevated above 18.9 GPa, it exhibited a blue shift instead, which was accompanied by considerable peak broadening and weakening of intensity.

At about 22.4 GPa, a shoulder peak (M_2_) appeared on the high wavenumber side of the O–H stretching modes and grew more obvious upon increasing pressure at the expense of the O–H stretching modes (Figure 5d). We calculated the bond lengths of the covalent O–H bonds in the monoclinic and orthorhombic phases. The bond lengths of the O–H bonds in the orthorhombic phase were slightly smaller than those in the monoclinic phase. This means that the stretching vibration frequency of O–H in the orthorhombic phase is higher than in the monoclinic phase. The correlation between the intensities of the O–H stretching modes of the initial phase and the new peak (M_2_) with pressure could be observed more clearly in the evolution of the Raman scattering spectra, as recorded in the decompression process. The M_2_ peak is unambiguous, with its intensity reduced during pressure decompression, and it can be retained at ambient pressure. At the same time, the intensities of the O–H stretching modes of the initial phase grow stronger. When the pressure is fully released, the peak M_2_ coexists with those of the initial monoclinic phase. According to group theory, the high-pressure phase belongs to the space group Pbn2_1_ (No. 33) and point group C_2v_ (mm^2^). The unit cell contains 64 atoms, hence there are 192 normal phonon modes at the Γ point:Γ_optic_ = 47A_1_ + 48A_2_ + 47B_1_ + 47B_2._(2)

The bending vibration modes of O–H partly disappeared at 20 GPa, and two new Raman peaks appeared at 167 cm^−1^ and 537 cm^−1^, which can be assigned to the vibration modes of the orthorhombic phase. The simultaneous presence of the bands of the monoclinic phase and the new bands characteristic of the orthorhombic phase over a long range of pressure suggests the coexistence of the monoclinic and high-pressure orthorhombic phases.

To gain further insight into the vibrational properties of Cu(OH)Cl, the pressure evolution of the Raman shifts was analyzed (Figure 7). The bending vibration modes and stretching vibration modes of O–H had obvious fluctuations when the pressure was higher than 14.7 GPa, and the changes of other vibration modes were not obvious. Above 21.7 GPa, the P dependence of Raman shifts seemed to show a bend at 21.7 GPa for the majority of Cu(OH)_3_Cl_3_ octahedra-frame modes. In the crystal structure of Cu(OH)Cl, a hydrogen bond (O–H···Cl) is formed between the Cl atom and the O–H group of the adjacent layers, respectively. At low pressures, the strength of the O–H···Cl hydrogen bond is enhanced with an increase in pressure, while the strength of the O–H covalent bond is decreased. The results show that the hydrogen bonding configuration of the initial phase changed at 14.7 GPa, and then the orthorhombic high-pressure phase appeared at 21.7 GPa. Combining the results of high-pressure XRD and Raman revealed that a new high-pressure phase appeared in Cu(OH)Cl at 18.7–21.7 GPa, which could coexist with the initial phase in a large pressure range, and the high-pressure phase could be partially retained in the material after the release of pressure.

In order to track the electron band gap evolution with pressure, UV/Vis absorption experiments were carried out on the Cu(OH)Cl samples under compression. Under ambient conditions, Cu(OH)Cl possesses an indirect-band gap of ∼2.4 eV. The valence band of Cu(OH)Cl is dominated by the Cl-2p orbital, while the conduction band is predominantly composed of the Cu-3d orbital (Figure 8a). At elevated pressures, the peculiar behavior of the bandgap of Cu(OH)Cl can be divided into three steps. First, the bandgap blueshifts steeply to about 2.5 eV when the pressure increased up to 7.2 GPa, with a mean pressure coefficient (dE_g_/dP) of 13 meV/GPa. Then a wide plateau, with a steady E_g_ of about 2.5 eV, dominates the curve between 7.2 and 17.9 GPa. Finally, as the pressure is elevated above 18.2 GPa, an abrupt decrease is observed in the bandgap, with significant bandgap narrowing from 2.48 eV to 2.27 eV at 26.1 GPa. However, when the pressure is further increased, detection of the optical transmission signal becomes difficult. At 18.2 GPa, a clear discontinuity in the bandgap evolution was evidenced, which is indicative of the possible structural changes at this pressure, confirming the occurrence of phase transition inferred from the RS and ADXRD studies.

Simulations based on the density functional theory (DFT) were undertaken to gain deeper insights into the electronic structure evolution under pressure. Figure 8b shows the calculated bandgaps versus pressure for the monoclinic and orthorhombic structures. The bandgap of the monoclinic structure increases firstly with pressure, reaching a maximum at about 8 GPa, and then decreases with pressure. The bandgap of the orthorhombic phase decreases monotonously with pressure, and the bandgap is narrower than that of the monoclinic phase at all pressures. Therefore, it can be speculated that the sudden decrease in band gap observed in the experiment may be related to the occurrence of phase transition from the monoclinic to the orthogonal structure.

The hydrogen bond by definition involves the lightest atom H and, hence, is very difficult to observe directly. The exact position of the hydrogen atoms cannot be directly determined from XRD experiments. In recent years, quantum mechanical calculations have been used increasingly to provide structural data (including but not restricted to the positions of hydrogen atoms) under a wide range of pressures. To gain further insight into the intrinsic nature involved in the compression behavior of the Cu(OH)Cl samples, density functional theory (DFT) calculations are performed to complement the experimental results and to understand the physical phenomena associated with the crystalline structure under high pressures. The calculated pressure dependencies of the interatomic (Cl···H, O···Cl, O–H) distances and the <O–H···Cl> bond angle in the P2_1_/a phase are shown in Figure 8c,d. The calculated Cl···H distance decreases monotonously when the pressure is lower than 16 GPa, and thereafter hardly changes with pressure. The calculated O···Cl distance decreases monotonously in the whole pressure range. At pressures below 12 GPa, the lengthening of the covalent O–H bond with increasing pressure is clearly visible. An inverse trend occurs as pressure is further elevated. The calculated <O–H···Cl> bond angle decreases almost linearly with increasing pressure in the whole range, with a kink occurring at about 16 GPa. Thus, it is clearly indicated that, under compression below 16 GPa, the O–H covalent bonds are weakened while the O–H···Cl hydrogen bonds are strengthened, which originates from a pressure-induced charge transfer from the region of covalent bonds to hydrogen bonds. When the pressure is above 16 GPa, the <O–H···Cl> bond angle decreases more rapidly, which means that the hydrogen bond tends to be weakened as the pressure increases. With the weakening of the hydrogen bond, the strength of the covalent bond O–H may be increased, leading to an inverse variation in the bond length of the O–H bond. These results are consistent with the variation in the Raman shift of the stretching vibration mode of the O–H bond, as observed experimentally. In addition, the hydrogen bonding in the high-pressure phase was investigated. Figure 9 gives the plots of the calculated geometric parameters versus pressure for the hydrogen bond configurations in the orthorhombic structure. At pressures higher than 20 GPa, the O–H bond length is considerably shorter, the H···Cl distance remains almost invariable, and the <O–H···Cl> bond angle is smaller, indicating that hydrogen bonding in the orthorhombic phase is weaker than that in the monoclinic phase.

At 16 GPa, exotic changes occur in the <O–H···Cl> bond angle and the covalent O–H bond length, which indicates that the initial structure may undergo a pressure-induced second-order phase transition. To verify this hypothesis, the phonon dispersion curves of Cu(OH)Cl in monoclinic structure have been calculated up to 24 GPa. As shown in Figure 10, there is no imaginary phonon frequency in the entire Brillouin zone, indicating that the monoclinic structure of Cu(OH)Cl is dynamically stable up to 24 GPa. In addition, the theoretical Raman-active mode frequencies of Cu(OH)Cl are calculated as a function of pressure up to 24 GPa. As shown in Figure 7b, the pressure evolution of the theoretical O–H stretching modes perfectly reproduces that observed in experiments, despite the fact that the calculated frequencies were slightly overestimated. This indicates that the theoretical simulation of the evolution of crystal structure with pressure is consistent with the experimental results. Thus, it may be concluded that the isostructural phase transition of the initial monoclinic phase occurs at the pressure of 16 GPa, due to the geometrical change in the hydrogen bond configuration.

Figure 11 shows the crystal structures of the monoclinic (M and M’) and orthorhombic (O) phases of Cu(OH)Cl involved in this study, with a schematic representation of the phase transition process. The symmetry before and after the isostructural phase transition (M to M’), which occurs at 16 GPa due to a change in the hydrogen bonding configuration, does not change, the difference between the two being a change in the relative positions between the different layers. The phase transition from monoclinic (M’) to orthorhombic (O) occurs at pressures above 18.7 GP, and the monoclinic and orthorhombic phases are able to coexist over a wide range of pressures. It is also notable that the basic building blocks of the monoclinic and orthorhombic structures are the same, the difference being that for the monoclinic phase, the stacking order between the layers is in the AAAA arrangement, whereas for the orthorhombic structure, the stacking order between the layers is the ABAB arrangement. Therefore, it can be concluded that the difference in the pressure-induced interlayer stacking order at 18.7 GPa leads to a structural phase transition from monoclinic to orthorhombic for Cu(OH)Cl.

## 4. Conclusions

The structure of a copper layered hydroxyhalogenide, belloite (Cu(OH)Cl), as a function of pressure, was studied using an in situ synchrotron X-ray, Raman scattering, and first-principle calculation. At high pressure, we observed the formation of hydrogen bonds between O–H∙∙∙Cl; the strength of the bonds increased and then decreased with increasing pressure. Cu(OH)Cl undergoes two transitions: an isostructural one at 14~16 GPa and a structural one at 18.7 GPa. The first transition was associated with layer sliding and a change in hydrogen bond configuration. The second transition induces an orthorhombic structure, which has a symmetry higher than P2_1_/a. Furthermore, we noticed that both the monoclinic and orthorhombic phases can coexist over a wide pressure range (18.7 to 28.4 GPa), which could be attributed to the fact that both phases possess a layered structure and are formed from similar basic units. The difference between these phases lies in the stacking sequence of the tectonic monolayers. In the monoclinic phase, they are arranged in an AAAA sequence. While in the orthorhombic phase, the monolayers are stacked in an ABAB sequence. Moreover, we found that the novel high-pressure phase (Phase-II) may be recovered to ambient conditions.

## Figures and Tables

**Figure 1 materials-14-05019-f001:**
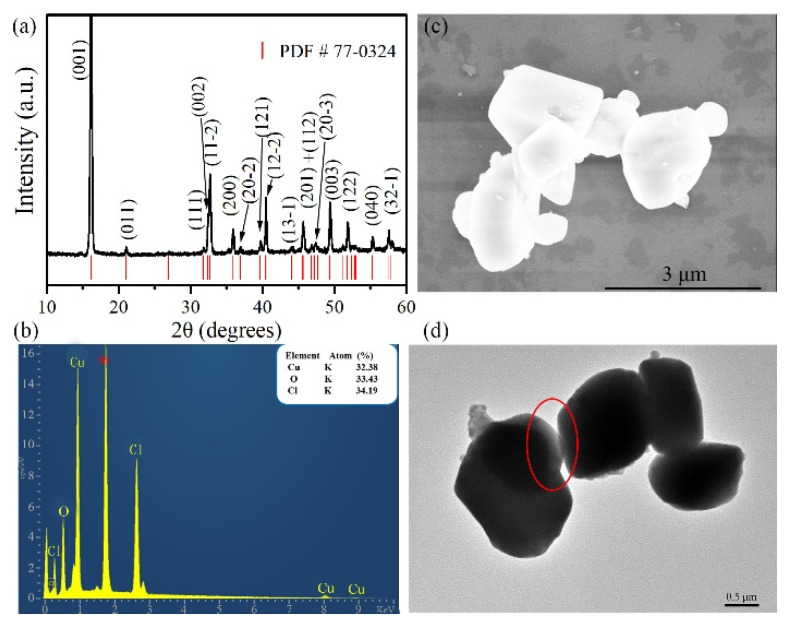
The typical XRD pattern (**a**), the EDX spectra (**b**), the SEM image (**c**), and the TEM image (**d**) of the prepared Cu(OH)Cl sample.

**Figure 2 materials-14-05019-f002:**
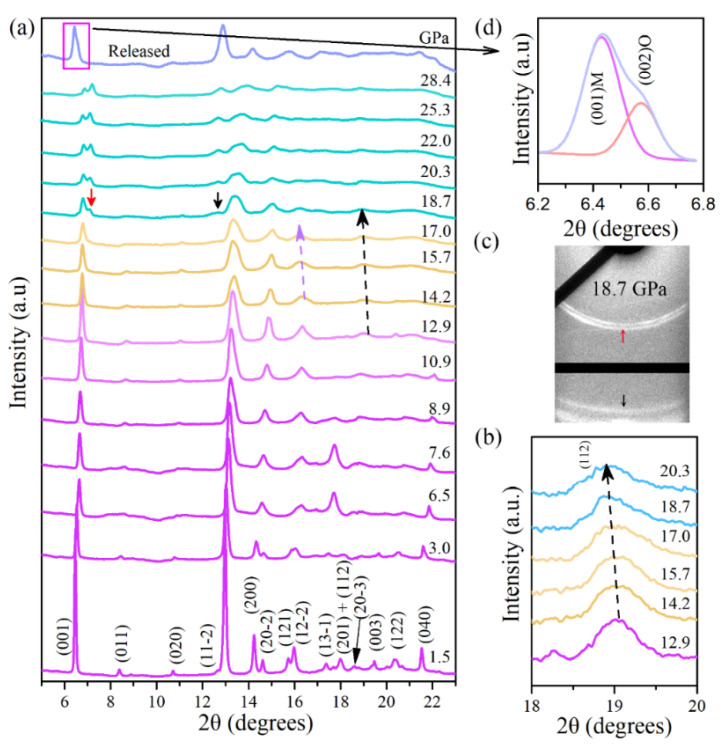
(**a**) The representative ADXRD patterns of Cu(OH)Cl measured at pressures up to 28.4 GPa at room temperature and released to ambient pressure (top pattern). The down arrows are used to mark the emergence of new peaks. (**b**) An enlarged view of the Bragg peak (1 1 2). (**c**) The two-dimensional ADXRD pattern of Cu(OH)Cl at 18.7 GPa. (**d**) The enlarged view of the split-peak fit near 6.2 Å after decompression.

**Figure 3 materials-14-05019-f003:**
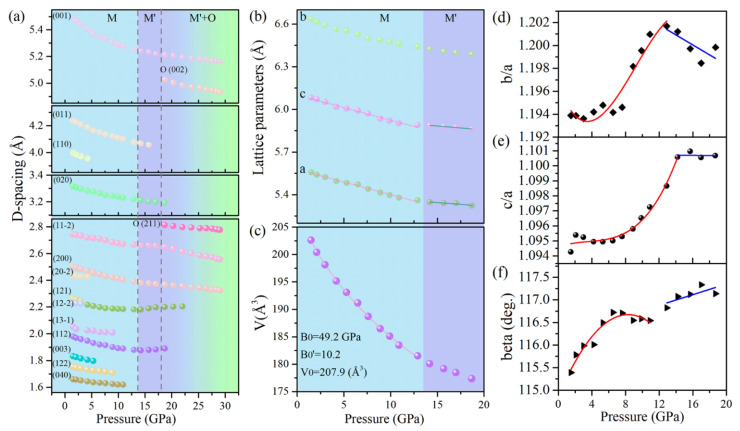
(**a**) The pressure-dependent d-spacings of Cu(OH)Cl. (**b**) The lattice parameters of Cu(OH)Cl as a function of pressure. (**c**) The pressure–volume relationship of Cu(OH)Cl. The spheres denote the experimental data points. They are fitted to the Birch–Murnaghan equation of state, as illustrated by the solid lines. (**d**) The evolution of the axial ratio b/a with pressure. (**e**) The evolution of the axial ratio c/a with pressure. (**f**) The evolution of the beta angle with pressure.

**Figure 4 materials-14-05019-f004:**
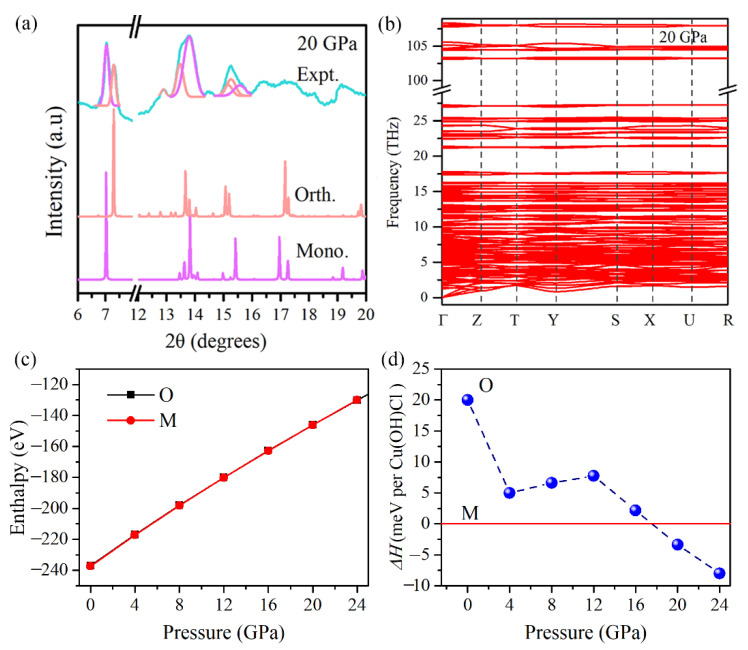
(**a**) The experimental and simulated XRD patterns of Cu(OH)Cl at 20 GPa with the two space groups, Pbn2_1_ and P2_1_/a. (**b**) Phonon dispersions of the Pbn2_1_ phase of Cu(OH)Cl at 20 GPa. (**c**) The calculated enthalpies of the initial monoclinic phase and the postulated high-pressure orthorhombic phase of Cu(OH)Cl. (**d**) The calculated relative enthalpies Δ*H* of the orthorhombic phase with respect to the monoclinic phase of Cu(OH)Cl under various pressures.

**Figure 5 materials-14-05019-f005:**
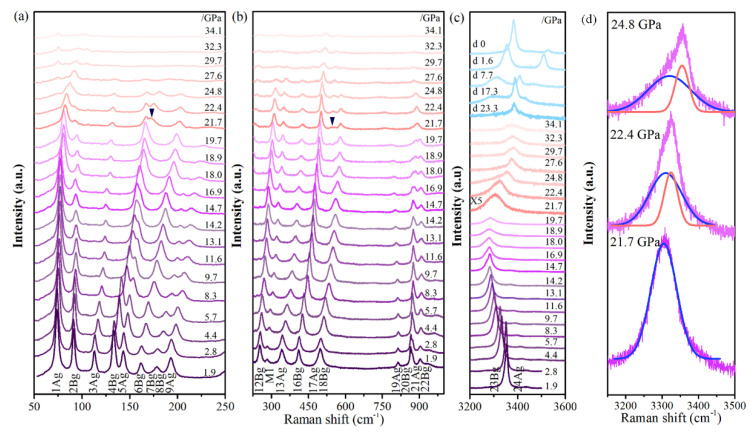
(**a**–**c**) Room-temperature Raman spectra of Cu(OH)Cl at selected pressures up to 34.1 GPa. (**d**) The enlarged picture of the Raman spectra at pressures of 21.7 GPa 22.4 GPa and 24.8 GPa.

**Figure 6 materials-14-05019-f006:**
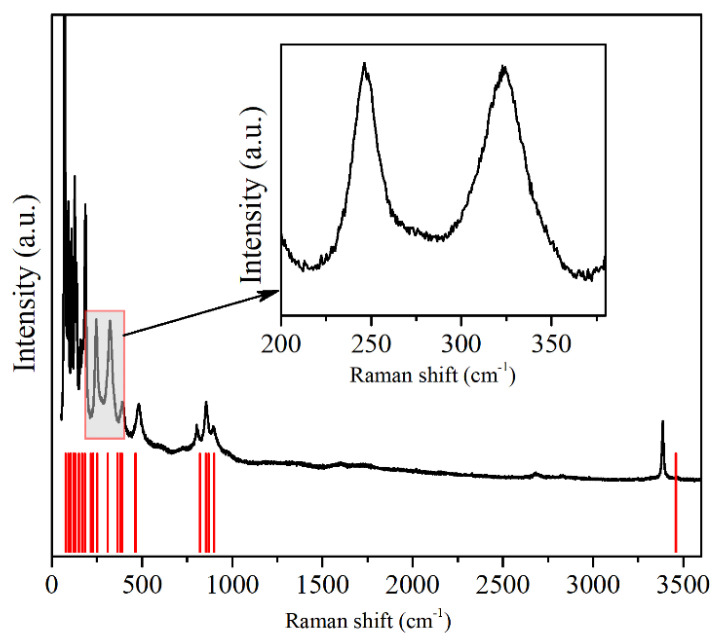
The typical Raman scattering spectra of Cu(OH)Cl at ambient pressure. The vertical lines represent the calculated locations of the Raman-active modes.

**Figure 7 materials-14-05019-f007:**
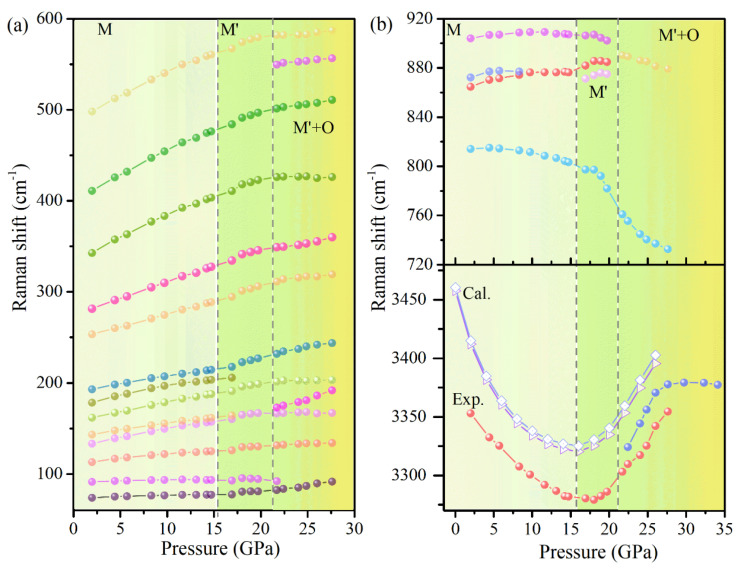
Experimental frequency versus pressure for the Raman active modes of Cu(OH)Cl and theoretical pressure dependence of the O–H stretching modes. (**a**) The lattice modes involving the deformation or relative displacement of the Cu(OH)_3_Cl_3_ octahedra. (**b**) Raman active bending and stretching modes of O–H group.

**Figure 8 materials-14-05019-f008:**
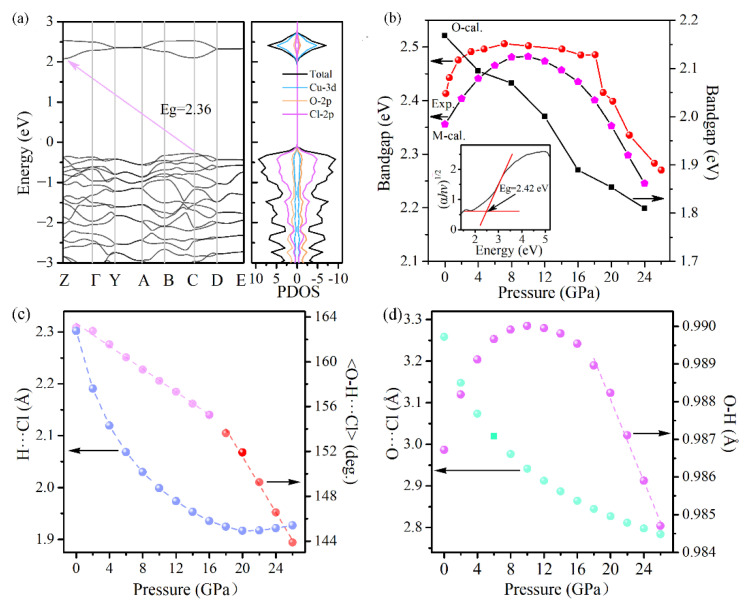
(**a**) The calculated electronic band structure and the partial density of states of Cu(OH)Cl at 0 GPa. (**b**) The experimental and calculated pressure dependences of the bandgaps of Cu(OH)Cl. The inset shows the linear range presented by the absorption edge of Cu(OH)Cl, corresponding to an indirect (α^1/2^) bandgap at ambient pressure. (**c**,**d**) The evolutions of the bond lengths with pressure for H−O, H···Cl, O−Cl, and the bond angle <O–H···Cl> in the monoclinic phase.

**Figure 9 materials-14-05019-f009:**
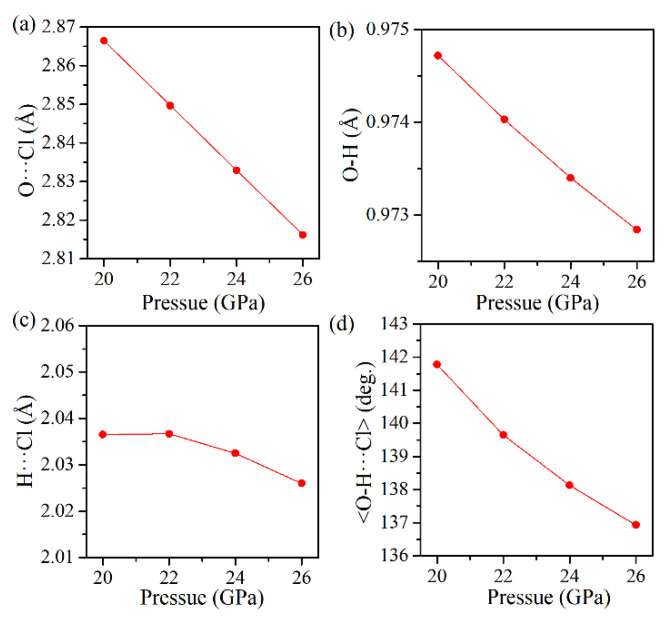
(**a**–**d**) The evolution of the calculated bond lengths with pressure for H−O, H···Cl, O−Cl, and the bond angles <O–H···Cl> in the orthorhombic phase.

**Figure 10 materials-14-05019-f010:**
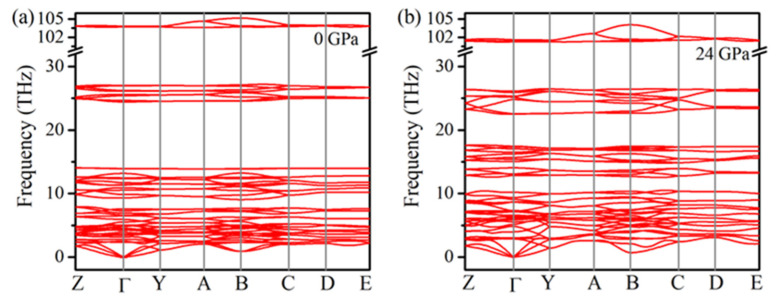
Phonon dispersions of the P2_1_/a phase of Cu(OH)Cl at selected pressures. (**a**) At 0 GPa. (**b**) At 24 GPa.

**Figure 11 materials-14-05019-f011:**
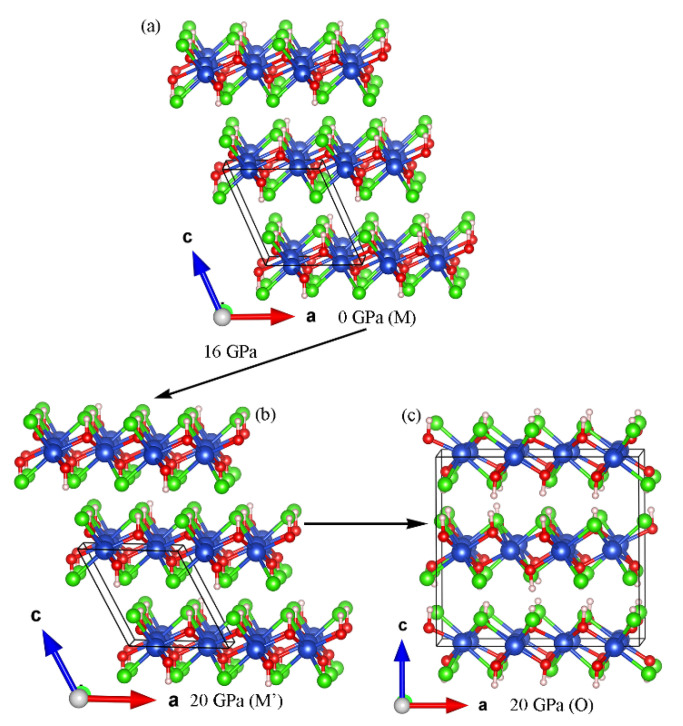
(**a**) and (**b**) show the crystal structures of Cu(OH)Cl in monoclinic at 0 GPa and 20 GPa, respectively. (**c**) shows the crystal structure of Cu(OH)Cl in orthorhombic phases at 20 GPa. The blue spheres represent Cu atoms, the red ones represent O atoms, the green ones represent Cl atoms, and the magenta ones represent H atoms.

**Table 1 materials-14-05019-t001:** Summarizes the theoretically calculated and the experimentally observed Raman-mode frequencies of the Cu(OH)Cl sample under study.

Modes	Cal.	Exp.	Modes	Calc.	Exp.	Modes	Cal.	Exp.
1A_g_	77	72	9A_g_	184	186	17A_g_	464	481
2B_g_	93	91	10B_g_	216		18B_g_	465	
3A_g_	105	109	11A_g_	230		19A_g_	822	807
4B_g_	120	127	12B_g_	253	247	20B_g_	854	856
5A_g_	130	137	13A_g_	311	324	21A_g_	870	880
6A_g_	151	158	14B_g_	363		22B_g_	899	897
7B_g_	167	171	15A_g_	378		23B_g_	3458	3388
8B_g_	168		16B_g_	390	391	24A_g_	3460	

## Data Availability

The data presented in this study are available on request from the corresponding author.

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
