# Peer review of "Pressure-Induced Variation of the Crystal Stacking Order in the Hydrogen-Bonded Quasi-Two-Dimensional Layered Material Cu(OH)Cl"

_materials, 2021, doi:10.3390/ma14175019_

Round 1

Reviewer 1 Report

In this paper the structural changes of Cu(OH)Cl with pressure is well studied showing a change of the stacking of the layers with increasing pressure.

This is a phenomenon known from other compounds, e.g. ZrNCl, but observed for the first time for Cu(OH)Cl.  As layered hydroxides and double hydroxides are of great importance for numerous applications such a model study at another example is important and can also be used for crystal growth of various modification or preparation of metastable variants.

The experiments are carefully done and well described. The interpretations are conclusive and the paper can be published in its present form.

Please check a little formats, e.g. P21/a (1 as subscript)...

Reviewer 2 Report

The manuscript “Pressure induced variation of the crystal stacking order in the hydrogen-bonded quasi-two-dimensional layered material Cu(OH)Cl” reports high-pressure structural phase transition and its mechanism on the quasi-two dimensional layer material Cu(OH)Cl. The authors carried out synchrotron radiation X-ray diffraction, optical Raman scattering, and optical absorption measurements in a diamond anvil cell (DAC), and found two-step pressure induced phase transition on the basis of changes of O-H stretching, O-H bending mode, and band gap under high pressures. By combining first-principles calculations, they confirmed that these phase transitions resulted from a changes of hydrogen bonding configuration and interlayer stacking order, respectively. The study of 2D layered materials, including low-dimensional van der Waals materials, is an important field for novel material properties and new functional materials, and high-pressure research on these materials is also expected to develop in the near future. I would recommend the ms to be published in Materials after revising the following questions and comments.

The authors used silicon oil and nitrogen as a pressure transmitting medium for XRD and optical absorption measurements and optical Raman spectroscopy, respectively. The silicon oil and nitrogen decrease hydrostaticity for pressures above 10 GPa, and then pressure gradients across a sample chamber become significant. This leads to a drastic decrease in the quality of the XRD and Raman data and in the accuracy of pressures. It seems that the quality of the XRD pattern at high pressure collected in this work is insufficient for quantitative analysis. This makes it impossible to identify the high pressure phase accurately. In addition, non-hydrostaticity seems to strongly affect the coexistence of the low-pressure and high-pressure phases under pressure. In this work, it is important to confirm the structural phase transition from monoclinic to orthorhombic structure, namely, a change of interlayer stacking order AAAA to ABAB. To do so, I propose that the authors should perform the DAC experiment using the best available pressure transmitting medium of helium and should collect high quality XRD data at high pressures.

The authors performed the first principle calculation of Cu(OH)Cl under high pressure conditions. However, no enthalpy calculation, which is useful to investigate thermodynamically structural stability, was shown on the low-pressure and high-pressure phases, including M' phase. Why not did the calculation report? Based on the present XRD data, it is difficult to determine the structure of the high pressure phase. Therefore, alternatively thermodynamically stable structural candidates should be proposed by theoretical calculations. If the high pressure phase were the orthorhombic structure, the length of hydrogen bonding and the bending angle should be calculated by the theoretical simulation and plotted in Fig. 6(c,d), together with those of the low pressure phase. Were the band gap on the low-pressure and high-pressure phases of Cu(OH)Cl not calculated under pressures?

Minor Comments
Line 81
The characterization of a starting material Cu(OH)Cl should be described, including homogeneity, size, and purity after sample synthesis using XRD and SEM, etc.

Line 96
No information of the culet size in the DAC.

Line 108
What power of excitation laser for Raman spectroscopy at sample position?
Could the intense laser damage the sample?

Fig. 2
The pressure dependence of the beta angle should be plotted, as one of lattice parameters. The axial ratio, c/a and/or b/a, should be also plotted to show a compression behavior of structure associated with the change of hydrogen bonding.

Line 176
The values of the bulk modulus and its pressure derivative are different between the text and Fig. 2. There is no discussion of the bulk modulus and its pressure derivative in comparison with similar two-dimensional layered materials.

Line 185
No citation of "Voronova et al." in the section of References.

Line 216
The values of Raman active mode frequency in Table 1 were at ambient pressure or not? If so, the Raman spectrum observed at ambient pressure should be shown, together with frequency positions of Raman active mode from the theoretical calculation. Why not were some Raman modes, e.g., 10Bg at 216 cm-1 and 11Ag at 230cm-1, observed in spite of theoretically prediction.

Fig. 8
The label of atoms, including Cu, O, H, and Cl, must be described in Figure or its caption. For reader, the authors should illustrate how the hydrogen atoms change their configuration when the M structure changes to the M' one.

The style of superscript and subscript should be checked in the text body and Table.

Reviewer 3 Report

In this manuscript by Hui Tian et al., the authors report the pressure control of stacking order and structural phase transition in hydrogen-bonded quasi-two-dimensional layered material Cu(OH)Cl through the combination of experimental measurements and first-principles calculations. They find that a structural phase transition from monoclinic phase to orthorhombic phase takes place  at pressure of 18.7 GPa. Overall, the present work shows some interesting results, relevant for the field of 2D materials, pressure control of symmetry and phase transition. If the authors can address the following comments, I can recommend it be published in Materials.

1: The authors should show some morphological characterization of the material Cu(OH)Cl, e.g., optical image, scanning electron microscope and transmission electron microscope.

2: The authors should give the full names of many abbreviations of terms, such as “DAC”, “ADXRD” and “DFT”. Please check the entire article carefully.

3: In line 134, the authors write “the Bragg peaks of (1 1 2) and (1 2 -2) shifted evidently to lower angles”. For Bragg peak (1 1 2), I cannot see it clearly since its intensity is too weak. Can the authors show an enlarged view?

4: The authors write that “At 1.9 GPa, a new Raman 234 mode at 290 cm− 1 (M1) appears and increases” (line 234). Can the authors show the Raman spectra at 0 GPa. Without the Raman spectra at 0 GPa, how do we know this Raman mode is a new one?

5: Besides, I would advise the authors to better review the existing literature that is of relevance for the present work. For example, stacking order and crystal phase control are effective structural approaches to engineer the symmetry breaking [Nat. Rev. Phys. 3, 193-206 (2021)]. For example, for bilayer transition metal dichalcogenides, 3R stacking lacks inversion symmetry, while 2H stacking has the inversion symmetry. Because of the broken inversion symmetry, bilayer transition metal dichalcogenides with 3R stacking can show exotic valley physics and nonlinear optical responses [Nat. Nanotechnol. 9, 825-829 (2014), Nat. Rev. Phys. 3, 193-206 (2021), Research 2019, 6494565 (2019), Phys. Rev. B 100, 161404 (2019), ACS Nano 8, 2951-2958 (2014), Light Sci. Appl. 5, e16131 (2016)]. As symmetry breaking is a very interesting topic, the authors may can show some discussions about this.

6: There are some errors. For example, “1 × 10−3 eV Å−1” (line 120) should be changed to “1 × 10−3 eV Å−1”; “CuOHF” (line 147) should be changed to “Cu(OH)Cl”; “point group C52h” (line 210) should be changed to “point group C52h”. Please check the entire article carefully.

Round 2

Reviewer 2 Report

The revised manuscript has been improved, and the reviewers' comments are reflected in the revised ms. I recommend this paper to be published in Materials after the minor revision.

Typos:
Line 152, Figure 1(d) instead of Figure 2(d).
Line 174, arrows instead of asterisks. No asterisks in Fig.2(a) are shown.

Fig. 1(a)
No information of X-rays wavelength. Which was Synchrotron or laboratory sources?
